# Strategic Behavior is Bliss: Iterative Voting Improves Social Welfare

**Joshua Kavner**
Department of Computer Science
Rensselaer Polytechnic Institute
Troy, NY 12180
kavnej@rpi.edu

**Lirong Xia**
Department of Computer Science
Rensselaer Polytechnic Institute
Troy, NY 12180
xialirong@gmail.com

## Abstract

Recent work in iterative voting has defined the additive dynamic price of anarchy (ADPoA) as the difference in social welfare between the truthful and worst-case equilibrium profiles resulting from repeated strategic manipulations. While iterative plurality has been shown to only return alternatives with at most one less initial votes than the truthful winner, it is less understood how agents' welfare changes in equilibrium. To this end, we differentiate agents' utility from their manipulation mechanism and determine iterative plurality's ADPoA in the worst- and average-cases. We first prove that the worst-case ADPoA is linear in the number of agents. To overcome this negative result, we study the average-case ADPoA and prove that equilibrium winners have a constant order welfare advantage over the truthful winner in expectation. Our positive results illustrate the prospect for social welfare to increase due to strategic manipulation.

## 1 Introduction

Voting is one of the most popular methods for a group of agents to make a collective decision based on their preferences. Whether a decision is for a high-stakes presidential election or a routine luncheon, agents submit their preferences and a voting rule is applied to select a winning alternative.

One critical flaw of voting is its susceptibility to strategic manipulations. That is, agents may have an incentive to misreport their preferences (i.e. votes) to obtain a more favorable outcome. Unfortunately, manipulation is inevitable under any non-dictatorial single-round voting systems when there are three or more alternatives, as recognized by the celebrated Gibbard-Satterthwaite theorem [Gibbard, 1973, Satterthwaite, 1975]. Consequently, decades of research sought to deter manipulation, especially by high computational barriers [Bartholdi et al., 1989, Faliszewski et al., 2010, Faliszewski and Procaccia, 2010]; see [Conitzer and Walsh, 2016] for a recent survey of the field.

While there is a large body of literature on manipulation of single-round voting systems, sequential and iterative voting procedures are less understood. Indeed, these procedures occur in a variety of applications, such as Doodle or presidential election polls, where people finalize their votes after previewing others' responses [Meir et al., 2010, Desmedt and Elkind, 2010, Xia and Conitzer, 2010, Reijngoud and Endriss, 2012, Zou et al., 2015]. Our key question is:

*What is the effect of strategic behavior in sequential and iterative voting?*

A series of work initiated by Meir et al. [2010] characterizes the dynamics and equilibria of *iterative voting*, where agents sequentially and myopically improve their reported preferences based on other agents' reports [Reyhani and Wilson, 2012, Lev and Rosenschein, 2012, Brânzei et al., 2013, Grandi et al., 2013, Obraztsova et al., 2013, Meir et al., 2014, Rabinovich et al., 2015, Obraztsova et al.,

35th Conference on Neural Information Processing Systems (NeurIPS 2021).

2015, Endriss et al., 2016, Meir, 2016, Tsang and Larson, 2016, Koolyk et al., 2017]. While the convergence of iterative voting has been investigated for many commonly studied voting rules, the effect of strategic behavior, in terms of aggregate social welfare, remains largely unclear.

A notable exception is Brânzei et al. [2013]'s work that introduced and characterized the *additive dynamic price of anarchy* (ADPoA) of iterative voting with respect to the plurality, veto, and Borda social choice functions. The (additive) DPoA measures the social welfare (difference) ratio between the truthful winner and an iterative policy's equilibrium winners when an adversary minimizes aggregate social welfare by controlling both the order in which agents make their strategic manipulations and agents' truthful preferences altogether. In particular, Brânzei et al. [2013] proved that under iterative plurality, the number of agents whose top preference is an equilibrium winner is at most one less than that of the truthful plurality winner. Therefore, strategic behavior does not have a significant negative impact on the social welfare measured by the sum plurality score of the winner. Nevertheless, it is unclear whether this observation holds for other notions of social welfare.

## 1.1 Our Contributions

We address the key question discussed above in the iterative voting framework, first proposed by Meir et al. [2010], by characterizing Brânzei et al. [2013]'s ADPoA under plurality dynamics and *rank-based utility functions* that differ from the iteration method. Given $m \geq 3$ alternatives, a ranked-based utility function is characterized by a utility vector $\vec{u}$ such that each agent receives $u_i$ utility if their $i$-th ranked alternative wins, although this alternative may differ for each agent. We study iterative plurality due to its simplicity and popularity in practice. Moreover, our results absolve the need for the mechanism's center to know $\vec{u}$ exactly, thus conserving agents' privacy. Still, we assume this is constant for all agents.

Our first main result (Theorem 1) states that, unfortunately, for any fixed $m \geq 3$ and utility vector $\vec{u}$, the ADPoA is $\Theta(n)$ for $n$ agents. Therefore, the positive result achieved by Brânzei et al. [2013] is not upheld if $\vec{u}$ differs from plurality utility under the iterative plurality mechanism.

To overcome this negative worst-case result, we introduce the notion of *expected additive dynamic price of anarchy* (EADPoA), which presumes agents' truthful preferences to be generated from a probability distribution. Our second main result (Theorem 2) is positive and surprises us: for any fixed $m \geq 3$ and utility vector $\vec{u}$, the EADPoA is $-\Omega(1)$ when agents' preferences are i.i.d. uniformly at random, known as *Impartial Culture (IC)* in social choice. In particular, our result suggests that *strategic behavior is bliss* because iterative voting helps agents choose an alternative with higher expected social welfare, regardless of the order of agents' strategic manipulations.

**Techniques.** We compute the EADPoA by partitioning the (randomly generated) profiles according to their *potential winners* – the alternatives that can be made to win by incrementing their plurality scores by at most one. Conditioned on profiles with two potential winners, we show that iterative plurality returns the alternative that beats the other in a head-to-head competition (Lemma 1). This type of "self selection" improves the expected social welfare over truthful plurality winner by $\Omega(1)$ (Lemma 2). When there are three or more potential winners, we further show that the expected welfare loss is $o(1)$ (Lemmas 3 and 4). Since the likelihood of $k$-way ties is exponentially small (in fact, $\Theta\left(n^{-\frac{k-1}{2}}\right)$ [Xia, 2021]), the overall social welfare is improved in expectation. We provide an experimental justification of our second main result in Appendix B.

## 1.2 Related Work and Discussions

**Sequential and iterative voting.** Since iterative voting's inception in 2010, many researchers have studied its convergence properties under differing assumptions and iteration protocols. Meir et al. [2010] first established the convergence of iterative plurality with deterministic tie-breaking from any initial preference profile or with randomized tie-breaking from the truthful profile. However, this result appears quite sensitive to its assumptions, since the authors found counter-examples when allowing agents to manipulate simultaneously, using better- instead of best-replies, or weighing agents' votes unequally. Lev and Rosenschein [2012] and Reyhani and Wilson [2012] independently showed that no other scoring rule besides veto necessarily converges, while Koolyk et al. [2017] demonstrated the same for common non-scoring rules, such as Maximin, Copeland, and Bucklin.

Similarly, Obraztsova et al. [2013, 2015] and Rabinovich et al. [2015] each analyze the conditions for Nash equilibrium for iterative voting rules and their truth-biased or lazy voting counterparts [Thompson et al., 2013]. Rabinovich et al. [2015] conclude that determining whether a given profile is a reachable Nash equilibrium is NP-complete.

Most iterative voting rules require agents to have full information about each others' votes in order to compute their best-responses. To relax this strong assumption, Reijngoud and Endriss [2012] and Endriss et al. [2016], inspired by Chopra et al. [2004], introduce communication graphs and poll information functions that restricts the amount information each agent receives and better respects voter privacy. The researchers subsequently provide susceptibility, immunity, and convergence results according to different voting rules. Tsang and Larson [2016] use these concepts to simulate iterative plurality on a social network and allow agents to infer their best-responses based on their neighbors' reports. The authors demonstrate how correlating agents' preferences affects the PoA and DPoA of strategic outcomes.

Sequential but non-iterative voting games have also been investigated in the literature. Desmedt and Elkind [2010] characterized the subgame perfect Nash equilibrium of a voting game where agents vote sequentially and are allowed to abstain from voting. Xia and Conitzer [2010] characterized the subgame perfect Nash equilibrium of a similar voting game where agents are not allowed to abstain, and proved that the equilibrium winner is highly unfavorable in the worst case, which can be viewed as an ordinal PoA. Our paper focuses on iterative voting setting proposed by Meir et al. [2010] and therefore differs from these works.

**Best-response mechanisms.** A separate line of research from iterative voting has studied the convergence and acyclicity properties of best-response mechanisms. Monderer and Shapley [1996] first introduced the finite improvement property applied to games with agents that sequentially change their actions. Apt and Simon [2015] and Fabrikant et al. [2010] subsequently characterized better-response dynamics in weakly acyclic games, which encapsulate potential and dominance-solvable games, and demonstrate bounds on finding their Nash equilibrium. The relationship between iterative voting and best-response mechanisms was explored by Meir et al. [2014] and Meir [2016], who fully characterized the acyclicity and local dominance properties of iterative voting rules.

**Implications of our main results.** Our results provide completeness and explanatory power to the empirical studies of Grandi et al. [2013] and Tsang and Larson [2016]. The former work shows an increase in additive social welfare using the Borda welfare vector due to plurality dynamics when agents have restricted manipulations and independent or correlated preferences [Berg, 1985]. The latter work shows a similar gain when agents have single-peaked preferences, are embedded on a social network, and make their manipulations based on estimates of their neighbors' reports. Put together, iterative voting provides a social welfare benefit that serves as an additional defense of strategic manipulation to those presented by Dowding and Hees [2008]. We believe our results will benefit a further study of non-strategyproof mechanisms in other social choice domains, such multi-issue voting [Bowman et al., 2014, Grandi et al., 2020].

## 2 Preliminaries

**Basic setting.** Let $\mathcal{A} = [m] \triangleq \{1, \ldots, m\}$ denote the set of $m \geq 3$ *alternatives* and $n \in \mathbb{N}$ denote the number of agents. We denote by $\mathcal{L}(\mathcal{A})$ the set of all strict linear orders over $\mathcal{A}$ and use $R_j \in \mathcal{L}(\mathcal{A}), j \leq n$ to represent agents' preference *rankings*. Preferences are aggregated into *profiles* $P = (R_1, \ldots, R_n)$, and we use $top(R_j) \in \mathcal{A}$ to denote agent $j$'s top preferred alternative. For any pair of alternatives $a, b \in \mathcal{A}$, we use $P[a \succ b]$ to denote the number of agents that prefer $a$ to $b$ in $P$.

**Integer positional scoring rules.** An *(integer) positional scoring rule* $r_{\vec{s}}$ is characterized by an integer scoring vector $\vec{s} = (s_1, \ldots, s_m) \in \mathbb{Z}_{\geq 0}^m$ with $s_1 \geq s_2 \geq \cdots \geq s_m \geq 0$ and $s_1 > s_m$. For example, *plurality* uses the vector $\vec{s}_{plu} = (1, 0, \ldots, 0)$, *veto* uses $(1, \ldots, 1, 0)$, and *Borda* uses $(m-1, m-2, \ldots, 0)$. In this work we focus on the plurality rule $r_{plu} = r_{\vec{s}_{plu}}$ and define the *score* of $a \in \mathcal{A}$ according to profile $P$ as $s_P(a) = \sum_{R \in P} \mathbb{1}\{top(R) = a\}$. We use the resolute function $r_{plu}(P) = \arg\max_{a \in \mathcal{A}} s_P(a)$ to select a single winning alternative, breaking ties lexicographically and favoring that with the smallest index. Assume $r = r_{plu}$ unless stated otherwise.

**Rank-based utility and additive social welfare.** We assume that agents have additive utilities characterized by a *rank-based utility vector* $\vec{u} = (u_1, \ldots, u_m) \in \mathbb{R}^m_{\geq 0}$ with $u_1 \geq \ldots \geq u_m \geq 0$ and $u_1 > u_m$. Like the scoring rule, each agent $j$ gets $\vec{u}(R_j, a) = u_i$ utility for the alternative $a \in \mathcal{A}$ ranked $i^{th}$ in $R_j$. Unlike prior work, however, we do not presume that $\vec{u}$ is the same as the scoring vector $\vec{s}$. We define the additive *social welfare* of $a$ according to $P$ as $\text{SW}_{\vec{u}}(P, a) = \sum_{j=1}^n \vec{u}(R_j, a)$.

**Iterative plurality voting.** Given agents' truthful preferences $P$, we consider an iterative process of profiles $(P^t)_{t \geq 0}$ that describe agents' *reported preferences* (i.e. votes) $(R_1^t, \ldots, R_n^t)_{t \geq 0}$ over time. For each round $t$, one agent $j$ is chosen by a *scheduler* $\phi$ to make a myopic *improvement step*, denoted by $R_j^t \xrightarrow{j} R_j'$, to their report. This step is called a *better-response* if $j$ prefers the new outcome $r(R_j', R_{-j}^t)$ to the prior one $r(R_j^t, R_{-j}^t)$, whereas it is a *best-response* (BR) if, additionally, $j$ could not have achieved a more preferred outcome than manipulating to $R_j'$ from $P^t$. [1]

Following Brânzei et al. [2013], we limit our discussion to strategic manipulations beginning from the truthful profile $P^0 = P$. This guarantees that all improvement steps $R_j' \xrightarrow{j} R_j''$ from profile $P'$ to $P''$ are best-responses that change the iterative winner: $r(P') \neq top(R_j') \wedge r(P'') = top(R_j'')$. [2] As a result, any sequence of BR steps converges in $\mathcal{O}(nm)$ rounds [Reyhani and Wilson, 2012]. The profiles $\{P^*\}$ with no further improvement steps are therefore *Nash equilibrium* (NE) with respect to $P$. We define $\text{EW}(P)$ as the set of *equilibrium winning* alternatives corresponding to all NE reachable from $P$ via some BR sequence. That is,

$$\text{EW}(P) = \{r(P^*) : \exists \text{ a BR sequence from } P \text{ leading to the NE profile } P^*\}$$

Lastly, we'll define the set of *potential winning* alternatives of any profile $P$ as those who could become a winner if their plurality score were to increment by one, including the current winner. That is, some agent could make these alternatives win by taking a BR step that increases their plurality score, if the agent's ranking permits. Following [Rabinovich et al., 2015], we have:

$$\text{PW}(P) = \left\{ a \in \mathcal{A} : \begin{cases} s_P(a) = s_P(r(P)) - 1, & a \text{ is ordered before } r(P) \\ s_P(a) = s_P(r(P)), & a \text{ is ordered after } r(P) \end{cases} \right\} \cup \{r(P)\}$$

where the ordering is lexicographical for tie-breaking. Reyhani and Wilson [2012] proved that the potential winning set is monotonic in $t$: $\forall t \geq 0$, $\text{PW}(P^{t+1}) \subseteq \text{PW}(P^t)$, which implies $\text{EW}(P) \subseteq \text{PW}(P^0)$. As a result, iterative plurality voting acts like a sequential tie-breaking mechanism whose outcome follows from the scheduler $\phi$. The following example demonstrates this section's concepts.

**Example 1.** *Let $n = 9$, $m = 3$, and consider the truthful profile $P$ defined with $R_1 = R_2 = R_3 = [1 \succ 3 \succ 2]$, $R_4 = R_5 = [2 \succ 3 \succ 1]$, $R_6 = [2 \succ 1 \succ 3]$, and $R_7 = R_8 = R_9 = [3 \succ 2 \succ 1]$. We observe from the plurality scores $(s_P(1), s_P(2), s_P(3)) = (3, 3, 3)$ that $r(P) = 1$ and $PW(P) = \{1, 2, 3\}$, representing a three-way tie. Next, Figure 1 describes the five BR sequences from $P$:*

Figure 1: Five BR sequences in Example 1. The tuples denote agents' reported top alternatives; the winner appears in curly brackets; arrows denote which agent makes each BR step and the updated report is emphasized.

*We therefore conclude $EW(P) = \{2, 3\}$. Moreover, consider the utility vector $\vec{u} = (u_1, u_2, u_3)$. Then the social welfare for each alternative is $(SW_{\vec{u}}(P, 1), \ SW_{\vec{u}}(P, 2), \ SW_{\vec{u}}(P, 3)) = (3u_1 + 1u_2 + 5u_3, \ 3u_1 + 3u_2 + 3u_3, \ 3u_1 + 5u_2 + 1u_3)$.* □

---

[1] Note that $\phi$ must select a BR step if one exists [Apt and Simon, 2015].

[2] These are characterized as *Type 1* or *direct best replies* in the literature [Meir, 2016]. Conversely, *Type 3* best-responses ($r(P') = top(R_j') \wedge r(P'') = top(R_j'')$) do not occur in improvement sequences from the truthful profile. Note that no BR step is of *Type 2* ($r(P') = top(R_j') \wedge r(P'') \neq top(R_j'')$).

# 3 Additive Dynamic PoA under General Utility Vectors

How bad are equilibrium outcomes, given that strategic manipulations inevitably occur by the Gibbard-Satterthwaite theorem [Gibbard, 1973, Satterthwaite, 1975]? Brânzei et al. [2013] sought to answer this question by defining the *additive dynamic price of anarchy (ADPoA)* as the *adversarial loss* – the difference in welfare between the truthful winner $r(P)$ and its worst-case equilibrium winner in $EW(P)$ – according to the worst-case $P$. To motivate this concept, consider users of a website that can regularly log in and update their preferences for an election. Then the ADPoA bounds the welfare loss if a virtual assistant can recommend when users should make their changes.

Brânzei et al. originally defined the ADPoA for a given positional scoring rule $r_{\vec{s}}$ and an additive social welfare function respecting $\vec{u} = \vec{s}$. In this case, the ADPoA of plurality was found to be 1, while the (multiplicative) DPoA of veto is $\Omega(m)$ and Borda is $\Omega(n)$ for $m \geq 4$ [Brânzei et al., 2013]. Although these results answer the authors' question and appear optimistic for plurality, they suggest more about the iteration mechanism than agents' collective welfare. For example, an ADPoA for plurality of 1 means that for any truthful profile, the difference in initial plurality scores of any equilibrium winner is at most one less that of the truthful winner. However, when we relax the utility vector $\vec{u}$ to differ from $\vec{s}$, we find in Theorem 1 that the ADPoA is quite poor at $\Theta(n)$.

First we recall Brânzei et al.'s definition of ADPoA using our notation and explicitly define the adversarial loss $D^+$ for a particular truthful profile $P$ before proceeding to our first main result.

**Definition 1** (**Additive Dynamic Price of Anarchy (ADPoA) [Brânzei et al., 2013]**). *Given a positional scoring rule $r_{\vec{s}}$, utility vector $\vec{u} = (u_1, \ldots, u_m)$ over $m \geq 3$ alternatives, and truthful profile $P$, the* adversarial loss starting from $P$ *is defined as*

$$D^+_{r_{\vec{s}}, \vec{u}}(P) = SW_{\vec{u}}(P, r_{\vec{s}}(P)) - \min_{a \in EW(P)} SW_{\vec{u}}(P, a)$$

*The* additive dynamic price of anarchy (ADPoA) *of $r_{\vec{s}}$ and $\vec{u}$ under $n$ agents is defined as*

$$ADPoA(r_{\vec{s}}, \vec{u}, n) = \max_{P \in \mathcal{L}(\mathcal{A})^n} D^+_{r_{\vec{s}}, \vec{u}}(P)$$

We will use ADPoA and $D^+$ to denote $ADPoA(r_{plu}, \vec{u}, n)$ and $D^+_{r_{plu}, \vec{u}}$ when the context is clear. [3] For example, we saw in Example 1 that $r(P) = 1$ and $EW(P) = \{2, 3\}$. Then

$$\begin{aligned}
D^+(P) &= \max\{ SW_{\vec{u}}(P, 1) - SW_{\vec{u}}(P, 2), \; SW_{\vec{u}}(P, 1) - SW_{\vec{u}}(P, 3) \} \\
&= \max\{ (3u_1 + 1u_2 + 5u_3) - (3u_1 + 3u_2 + 3u_3), \\
&\qquad\qquad (3u_1 + 1u_2 + 5u_3) - (3u_1 + 5u_2 + 1u_3) \} \\
&= -2(u_2 - u_3) \leq 0
\end{aligned}$$

Therefore the social welfare of both equilibrium winners is at least that of the truthful winner. In Theorem 2 below we'll see this conclusion hold in expectation. For the worst case profile $P$, however, the following theorem proves that this is not the case – rather, the worst-case equilibrium winner of $P$ has a social welfare linearly worse than the truthful winner.

**Theorem 1.** *Fix $m \geq 3$ and utility vector $\vec{u} = (u_1, \ldots, u_m)$. Then $ADPoA(r_{plu}, \vec{u}, n)$ is $\Theta(n)$. Specifically, $\forall n > 2m$,*

$$(u_2 - u_m)\left(\frac{n}{m} - 2\right) \leq ADPoA(r_{plu}, \vec{u}, n) \leq nu_1$$

*Proof.* The ADPoA is trivially upper bounded by the maximum social welfare attainable by any truthful profile $P$. For example, if $P$ is defined with $R_j = (1, 2, \ldots, m) \; \forall j \leq n$, then $\forall \tilde{P} \in \mathcal{L}(\mathcal{A})^n$,

$$D^+(\tilde{P}) = SW_{\vec{u}}(\tilde{P}, r(\tilde{P})) - \min_{a \in EW(\tilde{P})} SW_{\vec{u}}(\tilde{P}, a) \leq SW_{\vec{u}}(\tilde{P}, r(\tilde{P})) \leq SW_{\vec{u}}(P, r(P)) = nu_1$$

To lower bound ADPoA, we will construct a profile $P$ with a two-way tie between alternatives $1, 2 \in \mathcal{A}$ such that $D^+(P) = (u_2 - u_m)\left(\frac{n}{m} - 2\right)$. This implies the desired lower bound of

$$ADPoA = \max_{\tilde{P} \in \mathcal{L}(\mathcal{A})^n} D^+(\tilde{P}) \geq D^+(P) = (u_2 - u_m)\left(\frac{n}{m} - 2\right)$$

---

[3] The superscript '+' denotes an additive measure instead of multiplicative in the classical definition of PoA.

Fix $m \geq 3$ and let $n > 2m$ be even. We denote by $k = \arg\min_{\tilde{k} \in [2, m-1]}(u_{\tilde{k}} - u_{\tilde{k}+1})$ the position in $\vec{u}$ with the minimal difference in adjacent coordinates. Let $\alpha = \frac{1}{m}(n + m - 2)$ and $\beta = (\alpha - 1)(m - 2)$, such that $n = 2\alpha + \beta$. We will then construct $P$ as follows, with $\alpha$ agents that prefer 1 first and 2 last, $\alpha$ agents that prefer 2 first and 1 second, $(\frac{\beta}{2} - 1)$ agents that prefer 1 second and 2 last, and $(\frac{\beta}{2} + 1)$ agents that prefer 2 in their ranking's $k$-th position and 1 in their ranking's $(k+1)$-th position. We can see here that $s_P(1) = s_P(2) = \alpha$, and $\forall c > 2$, $s_P(c) = \alpha - 1$, thus guaranteeing the two-way tie. Therefore $r(P) = 1$ and $P[2 \succ 1] = \alpha + \frac{\beta}{2} + 1 > \alpha + \frac{\beta}{2} - 1 = P[1 \succ 2]$. This implies $\text{EW}(P) = \{2\}$ by the following lemma.

**Lemma 1.** *Let $m \geq 2$ and $a, b \in \mathcal{A}$ such that $a$ is ordered before $b$ in tie-breaking. Suppose $PW(P) = \{a, b\}$ for some truthful profile $P$. Then $EW(P) = \{a\}$ if $P[a \succ b] \geq P[b \succ a]$; otherwise $EW(P) = \{b\}$.*

The lemma's proof can be found in Appendix A.1. As a result,

$$D^+(P) = \text{SW}_{\vec{u}}(P, 1) - \text{SW}_{\vec{u}}(P, 2)$$

$$= \alpha(u_2 - u_m) + \left(\frac{\beta}{2} - 1\right)(u_2 - u_m) - \left(\frac{\beta}{2} + 1\right)(u_k - u_{k+1})$$

$$\geq (u_2 - u_m)(\alpha - 2) = (u_2 - u_m)\left(\frac{n - m - 2}{m}\right) \geq (u_2 - u_m)\left(\frac{n}{m} - 2\right)$$

where the first inequality holds because $(u_k - u_{k+1}) \leq (u_2 - u_m)$. $\qquad\square$

## 4 Expected Additive DPoA

In this section we extend Brânzei et al.'s ADPoA notion to account for the average-case adversarial loss for a positional scoring rule $r_{\vec{s}}$, rather than only the studying worst-case. This *expected additive dynamic price of anarchy* (EADPoA) bounds the adversarial loss of strategic manipulation according to more typical distributions of agents' rankings. Here we distribute profiles i.i.d. uniformly over $\mathcal{L}(\mathcal{A})^n$, known as the *Impartial Culture* distribution $\pi_n = IC^n$.

**Definition 2 (Expected Additive DPoA (EADPoA)).** *Given a positional scoring rule $r_{\vec{s}}$, a utility vector $\vec{u}$ over $m \geq 3$ alternatives, $n$ agents, and a distribution $\pi_n$ over $\mathcal{L}(\mathcal{A})^n$ for agents' preferences, the* expected additive dynamic price of anarchy *is defined as follows:*

$$EADPoA(r_{\vec{s}}, \vec{u}, n, \pi_n) = \mathbb{E}_{P \sim \pi_n}\left[D^+_{r_{\vec{s}}, \vec{u}}(P)\right]$$

Like before, we will use EADPoA and $D^+$ to denote $EADPoA(r_{plu}, \vec{u}, n, IC^n)$ and $D^+_{r_{plu}, \vec{u}}$ respectively. We similarly fix a rank-based utility vector $\vec{u}$ that may differ from the scoring rule $\vec{s}$, but we will not presume in the following theorem that this is known by the iterative plurality mechanism. In the subsequent proof, we will also drop the subscript "$P \sim IC^n$" to simplify notation when the context is clear.

**Theorem 2.** *Fix $m \geq 3$ and utility vector $\vec{u} = (u_1, \ldots, u_m)$. For any $n \in \mathbb{N}$ we have*

$$EADPoA(r_{plu}, \vec{u}, n, IC^n) = -\Omega(1)$$

*Proof.* The key idea is to partition $\mathcal{L}(\mathcal{A})^n$ according to each profile's potential winner set. More precisely, for every $W \subseteq \mathcal{A}$ with $W \neq \emptyset$, we define:

$$\overline{\text{PoA}}(W) = \Pr(PW(P) = W) \times \mathbb{E}[D^+(P) \mid PW(P) = W]$$

By the law of total expectation, then

$$\text{EADPoA} = \mathbb{E}[D^+(P)] = \sum_{\alpha=1}^{m} \sum_{W \subseteq \mathcal{A} : |W| = \alpha} \overline{\text{PoA}}(W) \tag{1}$$

where $\alpha$ denotes the number of potential winners in $P$. It is straightforward to see that when $\alpha = 1$, any profile $P$ with $|PW(P)| = 1$ is already a NE, which implies $D^+(P) = 0$. The rest of the proof proceeds as follows. For any $n \in \mathbb{N}$ we will show in Lemma 2 that for $\forall W \subseteq \mathcal{A}$ with $|W| = 2$

$\overline{\text{PoA}}(W) = -\Omega(1)$. We will then demonstrate that $\overline{\text{PoA}}(W) = o(1) \ \forall W \subseteq \mathcal{A}$ with $|W| = 3$ (Lemma 3) and $|W| \geq 4$ (if $m \geq 4$; Lemma 4). Recalling that $m$ is fixed, the total number of subsets of $\mathcal{A}$ is viewed as a constant. Finally, these results combine to conclude

$$\text{EADPoA} = \underbrace{0}_{\alpha=1} - \underbrace{\Omega(1)}_{\alpha=2} + \underbrace{o(1)}_{\alpha \geq 3} = -\Omega(1)$$

$\qquad\qquad\qquad\qquad\qquad\qquad\qquad\qquad\qquad\qquad\qquad\qquad\qquad\qquad\qquad\qquad\qquad\square$

Profiles with two tied alternatives drive the EADPoA negative because of the self-selecting property of Lemma 1. For example, consider a truthful $P$ with $\text{PW}(P) = \{a, b\}$ and $r(P) = a$. When more agents prefer the non-truthful winner $b$ in this setting, iterative plurality makes this correction by changing the winner to $b$ and increases agents' social welfare on average. When more agents prefer the truthful winner $a$, rather, iterative plurality doesn't change this outcome and the adversarial loss remains zero. Without a sufficient counter-balance to the former $\alpha = 2$ case by any of the $\alpha \geq 3$ cases, the adversarial loss overall remains negative in expectation.

The remainder of this section is devoted to detailing the proof behind the $\alpha = 2$ case (Lemma 2). We declare the $\alpha = 3$ case without proof (Lemma 3) and briefly prove the $\alpha \geq 4$ case (Lemma 4).

**Lemma 2 ($\alpha = 2$).** *Given $m \geq 3$ and a utility vector $\vec{u}$, for any $W \subseteq \mathcal{A}$ with $|W| = 2$ and any $n \in \mathbb{N}$, we have $\overline{PoA}(W) = -\Omega(1)$.*

*Proof.* Without loss of generality let $W = \{1, 2\}$ and suppose $u_2 > u_m$. There are two possible cases of $\text{PW}(P) = \{1, 2\}$: either $s_P(1) = s_P(2)$ or $s_P(1) + 1 = s_P(2)$, which we'll denote by $\mathcal{E}_1$ and $\mathcal{E}_2$ respectively. This suggests $\overline{\text{PoA}}(W) = \Pr(\mathcal{E}_1) \times \mathbb{E}[D^+(P) \mid \mathcal{E}_1] + \Pr(\mathcal{E}_2) \times \mathbb{E}[D^+(P) \mid \mathcal{E}_2]$. We'll focus on the first case where alternatives 1 and 2 are tied, since the latter's proof is similar.

We believe this proof is challenging due to the dependence in agents' rankings once we condition on profiles that satisfy two-way ties (i.e. $\mathcal{E}_1$). As a result, standard approximation techniques that assume independence, such as the Berry-Esseen inequality, no longer apply and may also be too coarse to support our claim. Instead, we will use a Bayesian network to further condition agents' rankings based on two properties: the top ranked-alternative and which of the two tied alternatives the agents prefer. Once we guarantee agents' rankings' conditional independence, we can identify the expected utility they gain for each alternative and then compute $\mathbb{E}[D^+(P) \mid \mathcal{E}_1]$ efficiently.

At a high level, there are two conditions for a profile $P$ to satisfy $\mathcal{E}_1$ and have non-zero adversarial loss. First, the profile must indeed be a two-way tie. This is represented in Step 1 below by identifying each agent $j$'s top-ranked alternative $t_j \in \mathcal{A}$ and conditioning $D^+(P)$ on a specific vector of top-ranked alternatives $\vec{t} \in \mathcal{T}_2 \subseteq \mathcal{A}^n$, a set corresponding to all profiles satisfying $\mathcal{E}_1$. Second, by Lemma 1, the profile should satisfy $P[2 \succ 1] \geq P[1 \succ 2]$. This is represented in Step 1 by identifying an indicator $z_j \in \{1, 2\}$ to suggest whether $1 \succ_j 2$ or $2 \succ_j 1$ respectively. We further condition $D^+(P)$ on a specific vector $\vec{z} \in \mathcal{Z}_{\vec{t},k}$, a set corresponding to all profiles in $\mathcal{E}_1$ with $k = P[2 \succ 1] \geq P[1 \succ 2] = n - k$. Once we condition $D^+(P)$ to satisfy these two conditions, we identify the expected difference in welfare between the alternatives $\mathbb{E}_{t_j,z_j}$ for each agent $j$ conditioned on $t_j, z_j$ in Step 2, which follows from the Impartial Culture assumption. Finally, we compute $D^+(P)$ by summing over all profiles satisfying the above two conditions and solve in Step 3, making use of Stirling's approximation.

More precisely, for any $j \leq n$, we represent agent $j$'s ranking distribution (i.i.d. uniform over $\mathcal{L}(\mathcal{A})$) by a Bayesian network of three random variables (see Figure 2). First, $T_j \in \mathcal{A}$ represents $j$'s top-ranked alternative and follows a uniform distribution. Second, $Z_j \in \{1, 2\}$ indicates whether $(1 \succ_j 2)$ or $(2 \succ_j 1)$ conditioned on $T_j$, and has probability $\{0.5, 0.5\}$ if $T_j \notin \{1, 2\}$. Third, $Q_j$ follows the uniform distribution over linear orders that uphold both $T_j$ and $Z_j$. It is not hard to verify that (unconditional) $Q_j$ follows the uniform distribution over $\mathcal{L}(\mathcal{A})$, which implies that $\vec{Q} = (Q_1, \ldots, Q_n)$ follows the same distribution as $P$.

**Step 1: Identify profiles that satisfy $\mathcal{E}_1$.** Let $\mathcal{T}_2 \subseteq [m]^n$ denote the set of top-ranked alternative vectors $\vec{t} = (t_1, \ldots, t_n)$ such that alternatives 1 and 2 have the maximum plurality score. Then $\mathcal{E}_1$ holds for $\vec{Q}$ if and only if $\vec{T}$ takes a value in $\mathcal{T}_2$.

$$\mathcal{T}_2 = \left\{ \vec{t} \in [m]^n : \forall 3 \leq i \leq m, \ |\{j : t_j = 1\}| = |\{j : t_j = 2\}| > |\{j : t_j = i\}| \right\}$$

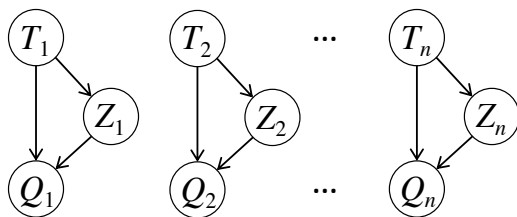

Figure 2: Bayesian network representation of $P$ as $\vec{T}$, $\vec{Z}$, and $\vec{Q}$ for the $\alpha = 2$ case.

Conditioned on agents' top-ranked alternatives being $\vec{t} \in \mathcal{T}_2$, we have by Lemma 1 that $D^+(\vec{Q})$ is non-zero if and only if $\vec{Q}[2 \succ 1] > \vec{Q}[1 \succ 2]$. Let $\text{Id}_1(\vec{t}) = \{j \leq n : t_j = 1\}$, $\text{Id}_2(\vec{t}) = \{j \leq n : t_j = 2\}$, and $\text{Id}_3(\vec{t}) = \{j \leq n : t_j \notin \{1, 2\}\}$ be the respective set of first-, second-, and third-party agents for $\vec{t}$. Since $\vec{t} \in \mathcal{T}_2$ implies $|\text{Id}_1(\vec{t})| = |\text{Id}_2(\vec{t})|$, there must be more third-party agents that prefer $2 \succ 1$ than those that prefer $1 \succ 2$. For every $\lceil \frac{|\text{Id}_3(\vec{t})|+1}{2} \rceil \leq k \leq |\text{Id}_3(\vec{t})|$, we thus define $\mathcal{Z}_{\vec{t},k} \subseteq \{1, 2\}^n$ as the set of all vectors $\vec{z}$ where the number of 2's in $\text{Id}_3(\vec{t})$ is exactly $k$.

$$\mathcal{Z}_{\vec{t},k} = \{\vec{z} \in \{1, 2\}^n : \forall j \in \text{Id}_1(\vec{t}) \cup \text{Id}_2(\vec{t}), z_j = t_j, \text{ and } |\{j \in \text{Id}_3(\vec{t}) : z_j = (2 \succ 1)\}| = k\}$$

By the law of total expectation and noting the independence of $\vec{Q}$'s components, we have

$$\Pr(\mathcal{E}_1) \times \mathbb{E}[D^+(P) \mid \mathcal{E}_1] = \sum_{\vec{t} \in \mathcal{T}} \sum_{k=\lceil \frac{|\text{Id}_3(\vec{t})|+1}{2} \rceil}^{|\text{Id}_3(\vec{t})|} \sum_{\vec{z} \in \mathcal{Z}_{\vec{t},k}} \Pr(\vec{T} = \vec{t}, \vec{Z} = \vec{z}) \sum_{j=1}^{n} E_{t_j, z_j} \tag{2}$$

where $E_{t_j, z_j} = \mathbb{E}_{\vec{Q}_j}[\vec{u}(Q_j, 1) - \vec{u}(Q_j, 2) \mid T_j = t_j, Z_j = z_j]$ is the expected difference in welfare between alternatives 1 and 2 for an agent $j$ with $T_j = t_j$ and $Z_j = z_j$.

**Step 2: Compute expected welfare difference per agent.** We note that $E_{t_j, z_j}$ only depends on the values of $t_j$ and $z_j$, but not $j$. The cases for $(t_j = z_j = 1)$ and $(t_j = z_j = 2)$ negate each other with $E_{1,1} + E_{2,2} = 0$. If $t_j \notin \{1, 2\}$ and $z_j = 1$, then $E_{t_j,1} = \eta > 0$ because $u_2 > u_m$. Similarly, it follows that if $t_j \notin \{1, 2\}$ and $z_j = 2$, then $E_{t_j,2} = -\eta$. Therefore Equation (2) becomes

$$\sum_{\vec{t} \in \mathcal{T}} \sum_{k=\lceil \frac{|\text{Id}_3(\vec{t})|+1}{2} \rceil}^{|\text{Id}_3(\vec{t})|} \sum_{\vec{z} \in \mathcal{Z}_{\vec{t},k}} \Pr(\vec{T} = \vec{t}, \vec{Z} = \vec{z}) \times (|\text{Id}_3(\vec{t})| - 2k)\eta \tag{3}$$

where we've inserted $\sum_{j=1}^{n} E_{t_j, z_j} = |\text{Id}_1(\vec{t})|E_{1,1} + |\text{Id}_2(\vec{t})|E_{2,2} - k\eta + (|\text{Id}_3(\vec{t})| - k)\eta$.

**Step 3: Simplify and solve.** Note that $\text{Id}_3(\vec{T})$ is equivalent to the sum of $n$ i.i.d. binary random variables, each of which is 1 with probability $\frac{m-2}{m} \geq \frac{1}{3}$. By Hoeffding's inequality, with exponentially small probability we have $\text{Id}_3(\vec{T}) < \frac{1}{6}n$. Therefore, we can focus on the $\text{Id}_3(\vec{T}) \geq \frac{1}{6}n$ case in (3), which, by denoting $\beta = |\text{Id}_3(\vec{t})|$ for ease of notation, becomes:

$$\leq e^{-\Omega(n)} + \sum_{\vec{t} \in \mathcal{T}_2 : \beta \geq \frac{1}{6}n} \sum_{k=\lceil \frac{\beta+1}{2} \rceil}^{\beta} \sum_{\vec{z} \in \mathcal{Z}_{\vec{t},k}} \Pr(\vec{T} = \vec{t}, \vec{Z} = \vec{z}) \times (\beta - 2k)\eta$$

$$= e^{-\Omega(n)} - \eta \sum_{\vec{t} \in \mathcal{T}_2 : \beta \geq \frac{1}{6}n} \left(\frac{1}{2}\right)^{\beta} \left(\left\lceil \frac{\beta+1}{2} \right\rceil\right) \binom{\beta}{\lceil \frac{\beta+1}{2} \rceil} \tag{4}$$

$$= e^{-\Omega(n)} - \eta \sum_{\vec{t} \in \mathcal{T}_2 : \beta \geq \frac{1}{6}n} \Pr(\vec{T} = \vec{t}) \times \Theta(\sqrt{n}) \tag{5}$$

$$= e^{-\Omega(n)} - \eta \Pr\left(\vec{T} \in \mathcal{T}_2, \text{Id}_3(\vec{T}) \geq \frac{1}{6}n\right) \times \Theta(\sqrt{n}) \leq -\Omega(1) \tag{6}$$

where Equation (4) follows from Claim 1 (see Appendix A.3) and Equation (5) follows from Stirling's approximation (see Appendix A.4). We get Equation (6) since $\Pr(\vec{T} \in \mathcal{T}_2)$ is equivalent to the probability of two-way ties under plurality w.r.t. IC, which is known to be $\Theta(n^{-1/2})$ [Gillett, 1977]. This concludes Lemma 2, and a more full proof can be found in Appendix A.2. □

**Lemma 3 ($\alpha = 3$).** *Given $m \geq 3$ and a utility vector $\vec{u}$, for any $W \subseteq \mathcal{A}$ with $|W| = 3$ and any $n \in \mathbb{N}$, we have $\overline{PoA}(W) = o(1)$.*

We defer the proof of Lemma 3 to Appendix A.5.

**Lemma 4 ($\alpha \geq 4$).** *Given $m \geq 4$ and a utility vector $\vec{u}$, for any $W \subseteq \mathcal{A}$ with $|W| \geq 4$ and any $n \in \mathbb{N}$, we have $\overline{PoA}(W) = o(1)$.*

*Proof.* The lemma follows after noticing the following. Firstly, we note that $\Pr(PW(P) = W) = \Theta(n^{-1.5})$ following a similar proof using the polyhedron representation as described in the proof of Lemma 3 (see Appendix A.5). Second, for any profile $P$, $D^+(P) = \mathcal{O}(n)$. □

## 5 Conclusions and Future Work

This paper studies the effects of strategic behavior in iterative plurality voting in terms of its adversarial loss – the difference in social welfare between the truthful winner and the worst-case equilibrium winning alternative. Our results naturally extend those of Brânzei et al. [2013] by utilizing rank-based utility functions whose utility vector $\vec{u}$ differs from the iterative positional scoring rule $r_{\vec{s}}$. We prove that iterative plurality has an adversarial loss linear in the number of agents in the worst case (Theorem 1). By distributing agents' preferences according to the impartial culture, we overcome this negative result and prove a constant order improvement in social welfare regardless of the order of agents' repeated strategic manipulations (Theorem 2). Even through our main result only works for IC, we are not aware of previous theoretical work on the expected performance of iterative voting under any distribution. Generalizing this study to other dynamics, utility functions, and families of distributions are interesting and important directions for future work.

For example, many iterative voting rules do not necessarily converge, but all games with best-response dynamics have cycles or steady-state equilibrium over agents' joint pure-strategy action space $\mathcal{L}(\mathcal{A})^n$ [Young, 1993, Meir, 2016]. Such games may instead be characterized by the worst-case ratio (or difference) between the social welfare of the game's truthful outcome and the average welfare of a stationary distribution over each cycle – known as the price of sinking [Goemans et al., 2005]. Bounding the welfare in each cycle could plausibly extend the DPoA, left for future work.

A second branch of future work could compare the iterative voting equilibrium winners' social welfare to that of the optimal winner, rather than the truthful outcome – known as the price of stability [Anshelevich et al., 2004, Tsang and Larson, 2016]. This is related to work in *distortion* which modifies agents' utility functions to be normalized [Procaccia and Rosenschein, 2006, Caragiannis and Procaccia, 2011] or embedded in a metric space [Anshelevich et al., 2018].

A third branch of future work could generalize the choice of agents' ranking distribution from IC, for example using smoothed analysis [Xia, 2020]. Determining the robustness of our theoretical results to other preference distributions, especially to those based on real-world data, would provide further insight into the effects of strategic manipulation on electoral outcomes. It would be interesting to see whether a greater proportion of i.i.d. preference distributions yield EADPoA results similar or dissimilar to that of IC.

## Acknowledgements

We thank anonymous reviewers for helpful comments. This work is supported by NSF #1453542, ONR #N00014-17-1-2621, and a giftfund from Google. J. Kavner acknowledges Abigail Jacobs for helpful discussions during the earliest stage of this work.

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
