# A  Deferred Proofs

## A.1  Proof of Lemma 1

**Lemma 1.** *Let $m \geq 2$ and $a, b \in \mathcal{A}$ such that $a$ is ordered before $b$ in tie-breaking. Suppose $\mathrm{PW}(P) = \{a, b\}$ for some truthful profile $P$. Then $\mathrm{EW}(P) = \{a\}$ if $P[a \succ b] \geq P[b \succ a]$; otherwise $\mathrm{EW}(P) = \{b\}$.*

*Proof.* Suppose $\mathrm{PW}(P) = \{a, b\}$ for some truthful profile $P$. First consider the case where $a$ and $b$ are tied with $s_P(a) = s_P(b)$. Let

- $\mathrm{Id}^{(a)}(P) = \{j \in [n] : top(R_j) \neq a, b, \text{ and } a \succ_j b\}$

- $\mathrm{Id}^{(b)}(P) = \{j \in [n] : top(R_j) \neq a, b, \text{ and } b \succ_j a\}$

denote the indices of agents who don't rank $a$ or $b$ highest but prefer $(a \succ b)$ or $(b \succ a)$ respectively. Since each BR sequence begins at $P^0 = P$, all BR steps are of Type 1 and must change the iterative winner each round, starting from $r(P^0) = a$. BR steps will therefore alternate whether they are taken by agents represented in $\mathrm{Id}^{(a)}(P)$ or $\mathrm{Id}^{(b)}(P)$. Agents from the former set will best-respond to rankings whose top preference is $a$, changing the winner to $a$, whereas agents from the latter set will best-respond to rankings whose top preference is $b$, changing the winner back to $b$. This alternation will continue until round $t$ when either $\mathrm{Id}^{(a)}(P^t)$ or $\mathrm{Id}^{(b)}(P^t)$ are emptied of indices. If $|\mathrm{Id}^{(a)}(P^0)| \geq |\mathrm{Id}^{(b)}(P^0)|$, the last BR step will make $a$ the unique equilibrium winner, whereas if $|\mathrm{Id}^{(a)}(P^0)| < |\mathrm{Id}^{(b)}(P^0)|$, the last BR step will make $b$ the unique equilibrium winner.

Inverse reasoning holds if $a$ and $b$ differ by one initial plurality score and $s_P(a) = s_P(b) - 1$, implying $r(P^0) = b$. In this case, the last BR step will make $a$ the unique equilibrium winner only if $|\mathrm{Id}^{(a)}(P^0)| > |\mathrm{Id}^{(b)}(P^0)|$, since the plurality score of $a$ is initially disadvantaged by 1. If not, the unique equilibrium winner will be $b$. We therefore conclude that if $P[a \succ b] \geq P[b \succ a]$ across all $n$ agents, then $\mathrm{EW}(P) = \{a\}$; otherwise $\mathrm{EW}(P) = \{b\}$.

$\square$

## A.2  Proof of Lemma 2

**Lemma 2 ($\alpha = 2$).** *Given $m \geq 3$ and a utility vector $\vec{u}$, for any $W \subseteq \mathcal{A}$ with $|W| = 2$ and any $n \in \mathbb{N}$, we have $\overline{PoA}(W) = -\Omega(1)$.*

*Proof.* Without loss of generality let $W = \{1, 2\}$ and suppose $u_2 > u_m$, since the case where $u_2 = u_m$ is covered in [Brânzei et al., 2013]. There are two possible cases of $\mathrm{PW}(P) = \{1, 2\}$: $\mathcal{E}_1 = \mathbb{1}\{s_P(1) = s_P(2)\}$, where 1 is the truthful winner, and $\mathcal{E}_2 = \mathbb{1}\{s_P(1) = s_P(2) - 1\}$, where 2 is the truthful winner. This suggests the following partition:

$$\overline{\mathrm{PoA}}(W) = \Pr(\mathcal{E}_1) \times \mathbb{E}[D^+(P) \mid \mathcal{E}_1] + \Pr(\mathcal{E}_2) \times \mathbb{E}[D^+(P) \mid \mathcal{E}_2]$$

We'll focus on the former summand where 1 and 2 are tied and prove that $\Pr(\mathcal{E}_1) \times \mathbb{E}[D^+(P) \mid \mathcal{E}_1] = -\Omega(1)$. The proof for the latter summand can be done similarly.

We believe this proof is challenging due to the dependence in agents' rankings once we condition on profiles that satisfy two-way ties (i.e. $\mathcal{E}_1$). As a result, standard approximation techniques that assume independence, such as the Berry-Esseen inequality, no longer apply and may also be too coarse to support our claim. Instead, we will use a Bayesian network to further condition agents' rankings based on two properties: the top ranked-alternative and which of the two tied alternatives the agents prefer. Once we guarantee agents' rankings' conditional independence, we can identify the expected utility they gain for each alternative and then compute $\mathbb{E}[D^+(P) \mid \mathcal{E}_1]$ efficiently.

At a high level, there are two conditions for a profile $P$ to satisfy $\mathcal{E}_1$ and have non-zero adversarial loss. First, the profile must indeed be a two-way tie. This is represented in Step 1 below by identifying each agent $j$'s top-ranked alternative $t_j \in \mathcal{A}$ and conditioning $D^+(P)$ on a specific vector of top-ranked alternatives $\vec{t} \in \mathcal{T}_2 \subseteq \mathcal{A}^n$, a set corresponding to all profiles satisfying $\mathcal{E}_1$. Second,

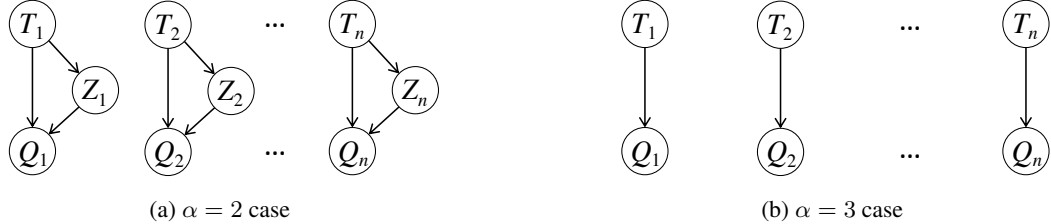

(a) $\alpha = 2$ case
(b) $\alpha = 3$ case

Figure 3: Bayesian network representation of $P$ as $\vec{T}$, $\vec{Z}$, and $\vec{Q}$

by Lemma 1, the profile should satisfy $P[2 \succ 1] \geq P[1 \succ 2]$. This is represented in Step 1 by identifying an indicator $z_j \in \{1, 2\}$ to suggest whether $1 \succ_j 2$ or $2 \succ_j 1$ respectively. We further condition $D^+(P)$ on a specific vector $\vec{z} \in \mathcal{Z}_{\vec{t}, k}$, a set corresponding to all profiles in $\mathcal{E}_1$ with $k = P[2 \succ 1] \geq P[1 \succ 2] = n - k$. Once we condition $D^+(P)$ to satisfy these two conditions, we identify the expected difference in welfare between the alternatives $\mathbb{E}_{t_j, z_j}$ for each agent $j$ conditioned on $t_j, z_j$ in Step 2, which follows from the Impartial Culture assumption. Finally, we compute $D^+(P)$ by summing over all profiles satisfying the above two conditions and solve in Step 3, making use of Stirling's approximation.

More precisely, for any $j \leq n$, we represent agent $j$'s ranking distribution (i.i.d. uniform over $\mathcal{L}(\mathcal{A})$) by a Bayesian network of three random variables: $T_j$ represents the top-ranked alternative, $Z_j$ represents whether $(1 \succ_j 2)$ or $(2 \succ_j 1)$, conditioned on $T_j$, and $Q_j$ represents the linear order conditioned on $T_j$ and $Z_j$. Formally, we have the following definition.

**Definition 3.** *For any $j \leq n$, we define a Bayesian network with three random variables $T_j \in \mathcal{A}$, $Z_j \in \{1, 2\}$, and $Q_j \in \mathcal{L}(\mathcal{A})$, where $T_j$ has no parent, $T_j$ is the parent of $Z_j$, and $T_j$ and $Z_j$ are $Q_j$'s parents (see Figure 3a). Let $\vec{T} = (T_1, , \ldots, T_n)$, $\vec{Z} = (Z_1, , \ldots, Z_n)$, and $\vec{Q} = (Q_1, , \ldots, Q_n)$. The (conditional) distributions are:*

- *$T_j$ follows a uniform distribution over $\mathcal{A}$*

- *$\Pr(Z_j = 1 \mid T_j = t) = \begin{cases} 1, & t = 1 \\ 0, & t = 2 \\ 0.5, & t \in [3, m] \end{cases}$*

- *$Q_j$ follows the uniform distribution over linear orders whose top alternative is $T_j$ and $(1 \succ_j 2)$ if $Z_j = 1$, or $(2 \succ_j 1)$ if $Z_j = 2$.*

It is not hard to verify that (unconditional) $Q_j$ follows the uniform distribution over $\mathcal{L}(\mathcal{A})$, which implies that $\vec{Q}$ follows the same distribution as $P$, namely $IC^n$. Notice that if alternative 1 or 2 is ranked at the top, then $Z_j$ is deterministic and equals to $T_j$. Furthermore, if $T_j \in \{1, 2\}$, then $Q_j$ follows the uniform distribution over $(m - 1)!$ linear orders; otherwise $Q_j$ follows the uniform distribution over $(m - 1)!/2$ linear orders.

**Example 2.** *Let $m = 4$ and $W = \{1, 2\}$. For every $j \leq n$, $T_j$ is the uniform distribution over $[4]$. We have that $\Pr(Z_j = 1 \mid T_j = 1) = \Pr(Z_j = 2 \mid T_j = 2) = 1$ and $\Pr(Z_j = 1 \mid T_j = 3) = \Pr(Z_j = 1 \mid T_j = 4) = 0.5$. Given $T_j = Z_j = 1$, $Q_j$ is the uniform distribution over*

$$\{[1 \succ 2 \succ 3 \succ 4], [1 \succ 2 \succ 4 \succ 3], [1 \succ 3 \succ 2 \succ 4]$$
$$[1 \succ 3 \succ 4 \succ 2], [1 \succ 4 \succ 2 \succ 3], [1 \succ 4 \succ 3 \succ 2]\}$$

*Given $T_j = 4$ and $Z_j = 2$, $Q_j$ is the uniform distribution over*

$$\{[4 \succ 2 \succ 1 \succ 3], [4 \succ 2 \succ 3 \succ 1], [4 \succ 3 \succ 2 \succ 1]\}$$

$\square$

**Step 1: Identify profiles that satisfy $\mathcal{E}_1$.** Let $\mathcal{T}_2 \subseteq [m]^n$ denote the set of vectors $\vec{t} = (t_1, \ldots, t_n)$ such that alternatives 1 and 2 have the maximum plurality score:

$$\mathcal{T}_2 = \{\vec{t} \in [m]^n : \forall 3 \leq i \leq m, |\{j : t_j = 1\}| = |\{j : t_j = 2\}| > |\{j : t_j = i\}|\}$$

$\mathcal{E}_1$ holds for $\vec{Q}$ if and only if $\vec{T}$ takes a value in $\mathcal{T}_2$, implying the following equality.

$$\Pr(\mathcal{E}_1) \times \mathbb{E}[D^+(P) \mid \mathcal{E}_1] = \sum_{\vec{t} \in \mathcal{T}_2} \Pr\left(\vec{T} = \vec{t}\right) \times \mathbb{E}_{\vec{Q}}[D^+(\vec{Q}) \mid \vec{T} = \vec{t}] \tag{7}$$

Conditioned on agents' top-ranked alternatives being $\vec{t} \in \mathcal{T}_2$, we have by Lemma 1 that $D^+(\vec{Q})$ is non-zero if and only if $\vec{Q}[2 \succ 1] > \vec{Q}[1 \succ 2]$ – thus $EW(\vec{Q}) = \{2\}$ is unique. For any $\vec{t} \in \mathcal{T}_2$, let

- $\text{Id}_1(\vec{t}) \subseteq [n]$ denote the indices $j$ such that $t_j = 1$

- $\text{Id}_2(\vec{t}) \subseteq [n]$ denote the indices $j$ such that $t_j = 2$

- $\text{Id}_3(\vec{t}) \subseteq [n]$ denote the indices $j$ such that $t_j \notin \{1, 2\}$ – we call these *third-party* agents

$\mathcal{E}_1$ implies $|\text{Id}_1(\vec{t})| = |\text{Id}_2(\vec{t})|$, so in order to uphold $\vec{Q}[2 \succ 1] > \vec{Q}[1 \succ 2]$ there must be more third-party agents that prefer $(2 \succ 1)$ than those that prefer $(1 \succ 2)$. Specifically, for every $\lceil \frac{|\text{Id}_3(\vec{t})|+1}{2} \rceil \leq k \leq |\text{Id}_3(\vec{t})|$, we define $\mathcal{Z}_{\vec{t},k} \subseteq \{1,2\}^n$ as the vectors $\vec{z}$ where the number of 2's among indices in $\text{Id}_3(\vec{t})$ is exactly $k$:

$$\mathcal{Z}_{\vec{t},k} = \{\vec{z} \in \{1,2\}^n : \forall j \in \text{Id}_1(\vec{t}) \cup \text{Id}_2(\vec{t}), z_j = t_j, \text{ and } |\{j \in \text{Id}_3(\vec{t}) : z_j = 2\}| = k\}$$

**Example 3.** *Suppose $m = 4$, $n = 9$, and $\vec{t} = (1,1,2,2,3,2,4,1,3)$. Then, $\text{Id}_1(\vec{t}) = \{1,2,8\}$, $\text{Id}_2(\vec{t}) = \{3,4,6\}$, $\text{Id}_3(\vec{t}) = \{5,7,9\}$. Moreover, for $k = 2$, we have*

$$\mathcal{Z}_{\vec{t},2} = \left\{ \begin{array}{l} (1,1,2,2,1,2,2,1,2) \\ (1,1,2,2,2,2,1,1,2) \\ (1,1,2,2,2,2,2,1,1) \end{array} \right\}$$

*where exactly two reports from agents 5, 7, or 9 are 2's: $|\{z_j = 2 : j \in \{5,7,9\}\}| = 2$.* $\qquad\square$

Continuing (7), we have

$$\Pr(\mathcal{E}_1) \times \mathbb{E}[D^+(P) \mid \mathcal{E}_1]$$

$$= \sum_{\vec{t} \in \mathcal{T}_2} \sum_{k=\lceil \frac{|\text{Id}_3(\vec{t})|+1}{2} \rceil}^{|\text{Id}_3(\vec{t})|} \sum_{\vec{z} \in \mathcal{Z}_{\vec{t},k}} \Pr(\vec{T} = \vec{t}, \vec{Z} = \vec{z}) \times \mathbb{E}_{\vec{Q}}[D^+(\vec{Q}) \mid \vec{T} = \vec{t}, \vec{Z} = \vec{z}]$$

$$= \sum_{\vec{t} \in \mathcal{T}_2} \sum_{k=\lceil \frac{|\text{Id}_3(\vec{t})|+1}{2} \rceil}^{|\text{Id}_3(\vec{t})|} \sum_{\vec{z} \in \mathcal{Z}_{\vec{t},k}} \Pr(\vec{T} = \vec{t}, \vec{Z} = \vec{z}) \sum_{j=1}^{n} \mathbb{E}_{\vec{Q}_j}[\vec{u}(Q_j, 1) - \vec{u}(Q_j, 2) \mid \vec{T} = \vec{t}, \vec{Z} = \vec{z}]$$

$$= \sum_{\vec{t} \in \mathcal{T}_2} \sum_{k=\lceil \frac{|\text{Id}_3(\vec{t})|+1}{2} \rceil}^{|\text{Id}_3(\vec{t})|} \sum_{\vec{z} \in \mathcal{Z}_{\vec{t},k}} \Pr(\vec{T} = \vec{t}, \vec{Z} = \vec{z}) \sum_{j=1}^{n} E_{t_j, z_j} \tag{8}$$

where

$$E_{t_j, z_j} = \mathbb{E}_{\vec{Q}_j}[\vec{u}(Q_j, 1) - \vec{u}(Q_j, 2) \mid T_j = t_j, Z_j = z_j]$$

The last equation holds because of the Bayesian network structure: for any $j \leq n$, given $T_j$ and $Z_j$, $Q_j$ is independent of other $Q$'s.

**Step 2: Computer expected welfare difference per agent.** $E_{t_j,z_j}$ only depends on the values of $t_j, z_j$ but not $j$:

- If $t_j = z_j = 1$, then $E_{t_j,z_j} = u_1 - \frac{u_2+\ldots+u_m}{m-1}$, the expected utility of alternative 2.

- If $t_j = z_j = 2$, then $E_{t_j,z_j}$ is the expected utility of alternative 1, which is $\frac{u_2+\ldots+u_m}{m-1}$, minus $u_1$. Notice that $E_{2,2} + E_{1,1} = 0$.

- If $t_j \notin \{1, 2\}$ and $z_j = 1$, then $\eta = E_{t_j,1}$ is the expected utility difference of alternatives 1 minus 2, conditioned on third-party agents and $(1 \succ 2)$. Note that $\eta > 0$ since $u_2 > u_m$.

- If $t_j \notin \{1, 2\}$ and $z_j = 2$, then $E_{t_j,2}$ is the expected utility difference of alternative 1 minus 2, conditioned on third-party agents and $(2 \succ 1)$. It follows that $E_{t_j,2} = -\eta$.

As a result, equation (8) becomes

$$\sum_{\vec{t}\in\mathcal{T}_2} \sum_{k=\lceil \frac{|\mathrm{Id}_3(\vec{t})|+1}{2}\rceil}^{|\mathrm{Id}_3(\vec{t})|} \sum_{\vec{z}\in\mathcal{Z}_{\vec{t},k}} \Pr(\vec{T} = \vec{t}, \vec{Z} = \vec{z}) \times (|\mathrm{Id}_3(\vec{t})| - 2k)\eta \tag{9}$$

where we've inserted

$$\sum_{j=1}^{n} E_{t_j,z_j} = |\mathrm{Id}_1(\vec{t})|E_{1,1} + |\mathrm{Id}_2(\vec{t})|E_{2,2} - k\eta + (|\mathrm{Id}_3(\vec{t})| - k)\eta$$

**Step 3: Simplify and solve.** Note that $\mathrm{Id}_3(\vec{T})$ is equivalent to the sum of $n$ i.i.d. binary random variables, each of which is 1 with probability $\frac{m-2}{m} \geq \frac{1}{3}$. By Hoeffding's inequality, with exponentially small probability we have $\mathrm{Id}_3(\vec{T}) < \frac{1}{6}n$. Therefore, we can focus on the $\mathrm{Id}_3(\vec{T}) \geq \frac{1}{6}n$ case in (9), which, by denoting $\beta = |\mathrm{Id}_3(\vec{t})|$ for ease of notation, becomes:

$$\leq \exp^{-\Omega(n)} + \sum_{\vec{t}\in\mathcal{T}_2:\beta\geq\frac{1}{6}n} \sum_{k=\lceil\frac{\beta+1}{2}\rceil}^{\beta} \sum_{\vec{z}\in\mathcal{Z}_{\vec{t},k}} \Pr(\vec{T} = \vec{t}, \vec{Z} = \vec{z}) \times (\beta - 2k)\eta$$

$$= \exp^{-\Omega(n)} + \sum_{\vec{t}\in\mathcal{T}_2:\beta\geq\frac{1}{6}n} \sum_{k=\lceil\frac{\beta+1}{2}\rceil}^{\beta} (\beta - 2k)\eta \sum_{\vec{z}\in\mathcal{Z}_{\vec{t},k}} \Pr(\vec{Z} = \vec{z} \mid \vec{T} = \vec{t}) \Pr(\vec{T} = \vec{t})$$

$$= \exp^{-\Omega(n)} + \sum_{\vec{t}\in\mathcal{T}_2:\beta\geq\frac{1}{6}n} \sum_{k=\lceil\frac{\beta+1}{2}\rceil}^{\beta} (\beta - 2k)\eta \left(\frac{1}{2}\right)^{\beta} \binom{\beta}{k} \Pr(\vec{T} = \vec{t}) \tag{10}$$

$$= \exp^{-\Omega(n)} + \sum_{\vec{t}\in\mathcal{T}_2:\beta\geq\frac{1}{6}n} \left(\frac{1}{2}\right)^{\beta} \eta \Pr(\vec{T} = \vec{t}) \sum_{k=\lceil\frac{\beta+1}{2}\rceil}^{\beta} \binom{\beta}{k}(\beta - 2k)$$

$$= \exp^{-\Omega(n)} - \eta \sum_{\vec{t}\in\mathcal{T}_2:\beta\geq\frac{1}{6}n} \left(\frac{1}{2}\right)^{\beta} \left(\left\lceil\frac{\beta+1}{2}\right\rceil\right) \binom{\beta}{\lceil\frac{\beta+1}{2}\rceil} \Pr(\vec{T} = \vec{t}) \tag{11}$$

where Equation (10) follows from $\Pr(Z_j = 1 \mid T_j \notin \{1, 2\}) = 0.5$ and Equation (11) follows from the following claim (Claim 1), plugging in $n \leftarrow \beta$ and $p \leftarrow \lceil\frac{\beta+1}{2}\rceil$.

**Claim 1.** *For any $n \in \mathbb{N}$ and any $p \in [0, n]$, we have*

$$\sum_{k=p}^{n} \binom{n}{k}(n - 2k) = -p\binom{n}{p}$$

The proof of Claim 1 can be found in Appendix A.3. We now apply Stirling's approximation to simplify Equation (11) as follows. See Appendix A.4, plugging in $u \leftarrow \beta$ which we recall is $\Theta(n)$.

$$e^{-\Omega(n)} - \eta \sum_{\vec{t} \in \mathcal{T}_2 : \beta \geq \frac{1}{6}n} \Pr(\vec{T} = \vec{t}) \times \Theta(\sqrt{n})$$

$$= e^{-\Omega(n)} - \eta \Pr\left(\vec{T} \in \mathcal{T}_2, \ \mathrm{Id}_3(\vec{T}) \geq \frac{1}{6}n\right) \times \Theta(\sqrt{n})$$

$$= e^{-\Omega(n)} - \eta \left(\Pr(\vec{T} \in \mathcal{T}_2) - \Pr\left(\vec{T} \in \mathcal{T}_2, \ \mathrm{Id}_3(\vec{T}) < \frac{1}{6}n\right)\right) \times \Theta(\sqrt{n})$$

$$\leq e^{-\Omega(n)} - \eta \left(\Pr(\vec{T} \in \mathcal{T}_2) - \Pr\left(\mathrm{Id}_3(\vec{T}) < \frac{1}{6}n\right)\right) \times \Theta(\sqrt{n})$$

$$\leq e^{-\Omega(n)} - \eta \left(\Theta(n^{-1/2}) - e^{-\Omega(n)}\right) \times \Theta(\sqrt{n})$$

$$= -\Omega(1)$$

where $\Pr(\vec{T} \in \mathcal{T}_2)$ is equivalent to the probability of two-way ties under plurality w.r.t. IC, which is known to be $\Theta(n^{-1/2})$ [Gillett, 1977]. This proves Lemma 2. $\qquad \square$

## A.3  Proof of Claim 1

**Claim 1.** *For any $n \in \mathbb{N}$ and any $p \in [0, n]$, we have*

$$\sum_{k=p}^{n} \binom{n}{k}(n - 2k) = -p\binom{n}{p}$$

*Proof.*

$$\sum_{k=p}^{n} \binom{n}{k}(n - 2k) = \sum_{k=0}^{n} \binom{n}{k}(n - 2k) - \sum_{k=0}^{p-1} \binom{n}{k}(n - 2k)$$

$$= n2^n - 2(n2^{n-1}) - \sum_{k=0}^{p-1} \binom{n}{k}(n - 2k)$$

$$= -\sum_{k=0}^{p-1} \binom{n}{k}(n - 2k)$$

The proof shall be continued by induction. We want to show that for all $p \in [0, n]$,

$$\sum_{k=0}^{p-1} \binom{n}{k}(n - 2k) = p\binom{n}{p} \tag{12}$$

**(Base step)** Substituting $p = 1$ into Equation (12) yields

$$\binom{n}{0}(n - 0) = n = 1\binom{n}{1}$$

**(Inductive step)** Suppose Equation (12) holds for all $p \in [0, n']$ for some $n' < n$. We want to show this holds for $p + 1$, or equivalently that:

$$\sum_{k=0}^{p} \binom{n}{k}(n - 2k) = p\binom{n}{p} + \binom{n}{p}(n - 2p) = (p + 1)\binom{n}{p+1}$$

where we've used the induction hypothesis in the first term's substitution. The middle term thus becomes

$$(n - p)\binom{n}{p} = \frac{n!(n - p)}{p!(n - p)!} = \frac{n!(p + 1)}{(p + 1)!(n - p - 1)!} = (p + 1)\binom{n}{p+1}$$

as desired. $\qquad \square$

## A.4 Application of Stirling's Approximation for Lemma 2

Let $u \in \mathbb{N}$ and set $v = \lfloor \frac{u}{2} \rfloor$. We can immediately see that $\lceil \frac{u+1}{2} \rceil = v + 1$, and from Equation (11) in Lemma 2, we want to simplify the term $(v + 1)\binom{u}{v+1}$. Stirling's approximation states that for every $n \in \mathbb{N}$,

$$n! \sim \sqrt{2\pi n} \left(\frac{n}{e}\right)^n$$

If $u$ is odd, then $u = 2v + 1$ and we have

$$\binom{u}{v+1}(v+1) = \frac{u!}{v!^2} \sim \frac{\left(\sqrt{2\pi u}u^u e^{-u}\right)}{\left(\sqrt{2\pi}v^{(v+0.5)}e^{-v}\right)^2} = \frac{\sqrt{u}}{\sqrt{2\pi}} \frac{(u^u e^{-u})}{\left(v^{(2v+1)}e^{-2v}\right)}$$

$$= \frac{\sqrt{u}}{e\sqrt{2\pi}} \left(\frac{u}{v}\right)^u = \frac{\sqrt{u}}{e\sqrt{2\pi}} \left(2 + \frac{1}{v}\right)^u$$

If $u$ is even, then $u = 2v$ and we have

$$\binom{u}{v+1}(v+1) = \frac{u!v}{v!^2} \sim \frac{\left(\sqrt{2\pi u}u^u e^{-u}\right)v}{\left(\sqrt{2\pi}v^{(v+0.5)}e^{-v}\right)^2} = \frac{\sqrt{u}}{\sqrt{2\pi}} \frac{(u^u e^{-u})v}{\left(v^{(2v+1)}e^{-2v}\right)} = \frac{\sqrt{u}2^u}{e\sqrt{2\pi}}$$

In both cases the objective scales as $\Theta(\sqrt{u}2^u)$.

## A.5 Proof of Lemma 3

**Lemma 3 ($\alpha = 3$).** *Given $m \geq 3$ and a utility vector $\vec{u}$, for any $W \subseteq \mathcal{A}$ with $|W| = 3$ and any $n \in \mathbb{N}$, we have $\overline{PoA}(W) = o(1)$.*

*Proof.* The proof uses a similar and simpler technique than that of Lemma 2. Without loss of generality, suppose $W = \{1, 2, 3\}$ and consider the case where the plurality scores for 1, 2, and 3 are equal, denoted $\mathcal{E}$. The proofs for cases with alternatives 2 or 3 being truthful winners are similar. We first prove that conditioned on the vector $\vec{t}$ of all agents' top preferences that satisfy $\mathcal{E}$, the maximum score difference between any pair of alternatives in $\{1, 2, 3\}$ is $o(n)$ with high probability that is close to 1. Secondly, we prove that $\text{PW}(P) = W$ with probability $\mathcal{O}(n^{-1})$.

More precisely, for every $j \leq n$, we represent agent $j$'s ranking distribution (i.i.d. uniform over $\mathcal{L}(\mathcal{A})$) by a Bayesian network of two random variables: $T_j$ represents agent $j$'s top-ranked alternative, and $Q_j$ represents $j$'s ranking conditioned on $T_j$. Formally, we have the following definition.

**Definition 4.** *For any $j \leq n$, we define a Bayesian network with two random variables $T_j \in \mathcal{A}$ and $Q_j \in \mathcal{L}(\mathcal{A})$, where $T_j$ has no parent and is the parent of $Q_j$ (see Figure 3b). Let $\vec{T} = (T_1, , \ldots, T_n)$ and $\vec{Q} = (Q_1, , \ldots, Q_n)$. The (conditional) distributions are:*

- *$T_j$ follows a uniform distribution over $\mathcal{A}$*
- *$Q_j$ follows the uniform distribution over linear orders whose top alternative is $T_j$*

It is not hard to verify that (unconditional) $Q_j$ follows the uniform distribution over $\mathcal{L}(\mathcal{A})$. Therefore, $\vec{Q}$ follows the same distribution as $P$, which is $\text{IC}^n$.

**Example 4.** *Let $m = 4$ and $W = \{1, 2, 3\}$. For every $j \leq n$, $T_j$ is the uniform distribution over $[4]$. Given $T_j = 1$, $Q_j$ is the uniform distribution over*

$$\{[1 \succ 2 \succ 3 \succ 4], [1 \succ 2 \succ 4 \succ 3], [1 \succ 3 \succ 2 \succ 4],$$
$$[1 \succ 3 \succ 4 \succ 2], [1 \succ 4 \succ 2 \succ 3], [1 \succ 4 \succ 3 \succ 2]\}$$

*Given $T_j = 4$, $Q_j$ is the uniform distribution over*

$$\{[4 \succ 1 \succ 2 \succ 3], [4 \succ 1 \succ 3 \succ 2], [4 \succ 3 \succ 1 \succ 2],$$
$$[4 \succ 2 \succ 1 \succ 3], [4 \succ 2 \succ 3 \succ 1], [4 \succ 3 \succ 2 \succ 1]\}$$

$\square$

**Step 1: Identify $\mathcal{E}$.** Let $\mathcal{T}_3 \subseteq [m]^n$ denote the set of vectors $\vec{t} = (t_1, \ldots, t_n)$ such that alternatives 1, 2, and 3 have the maximum plurality score. Formally,

$$\mathcal{T}_3 = \left\{ \vec{t} \in [m]^n : \forall 4 \leq i \leq m, |\{j : t_j = 1\}| = |\{j : t_j = 2\}| = |\{j : t_j = 3\}| > |\{j : t_j = i\}| \right\}$$

$\mathcal{E}$ holds for $\vec{Q}$ if and only if $\vec{T}$ takes a value in $\mathcal{T}_3$, implying the following equality.

$$\begin{aligned}
\overline{\text{PoA}}(\{1,2,3\}) &= \Pr\left( \text{PW}(\vec{Q}) = \{1,2,3\} \right) \times \mathbb{E}[\text{D}^+(\vec{Q}) \mid \text{PW}(\vec{Q}) = \{1,2,3\}] \\
&= \sum_{\vec{t} \in \mathcal{T}_3} \Pr(\vec{T} = \vec{t}) \times \mathbb{E}[\text{D}^+(\vec{Q}) \mid \vec{T} = \vec{t}]
\end{aligned} \tag{13}$$

**Step 2: Upper-bound the conditional adversarial loss.** We next employ the law of total expectation on Equation (13) by further conditioning on $\mathbb{1}\{D^+(\vec{Q}) > n^{0.6}\}$. This event represents whether the adversarial loss scales positively and at least sub-linearly in $n$. We will show this holds with high probability and establish the following conditional expectation to be $o(n)$, term-by-term:

$$\begin{aligned}
\mathbb{E}[\text{D}^+(\vec{Q}) \mid \vec{T} = \vec{t}] = \; &\mathbb{E}[\text{D}^+(\vec{Q}) \mid \vec{T} = \vec{t}, \text{D}^+(\vec{Q}) > n^{0.6}] \times \Pr(\text{D}^+(\vec{Q}) > n^{0.6} \mid \vec{T} = \vec{t}) \\
&+ \mathbb{E}[\text{D}^+(\vec{Q}) \mid \vec{T} = \vec{t}, \text{D}^+(\vec{Q}) \leq n^{0.6}] \times \Pr(\text{D}^+(\vec{Q}) \leq n^{0.6} \mid \vec{T} = \vec{t})
\end{aligned}$$

Trivially, we note that

$$\mathbb{E}[\text{D}^+(\vec{Q}) \mid \vec{T} = \vec{t}, \text{D}^+(\vec{Q}) \leq n^{0.6}] \leq n^{0.6} \tag{14}$$

Second, for any $t \in [m]$ and $i_1, i_2 \in \{1,2,3\}$ with $i_1 \neq i_2$, we denote by $D^t_{i_1, i_2}$ the random variable representing the utility difference between alternatives $i_1$ and $i_2$ in $Q_j$, conditioned on $T_j = t$:

$$D^t_{i_1, i_2} = \vec{u}(Q_j, i_1) - \vec{u}(Q_j, i_2)$$

Notice that $D^t_{i_1, i_2}$ does not depend on $j$. For any $\vec{t} \in [m]^n$ and $j \leq n$, $D^{t_j}_{i_1, i_2} \in [u_m - u_1, u_1 - u_m]$, which implies $\text{D}^+(\vec{Q}) \leq (u_1 - u_m)n$, and henceforth

$$\mathbb{E}[\text{D}^+(\vec{Q}) \mid \vec{T} = \vec{t}, \text{D}^+(\vec{Q}) > n^{0.6}] \leq (u_1 - u_m)n \tag{15}$$

Thirdly, we observe that $\mathbb{E}[D^{t_j}_{i_1, i_2}] > 0$ if $t_j = i_1$, $\mathbb{E}[D^{i_1}_{i_1, i_2}] = -\mathbb{E}[D^{i_1}_{i_1, i_2}] < 0$ if $t_j = i_2$, and $\mathbb{E}[D^{t_j}_{i_1, i_2}] = 0$ otherwise. Let $D^{\vec{t}}_{i_1, i_2} = \sum_{j=1}^n D^{t_j}_{i_1, i_2}$. It follows that for any $\vec{t} \in \mathcal{T}_3$ we have $\mathbb{E}[D^{\vec{t}}_{i_1, i_2}] = 0$, since $\mathcal{E}$ implies $|\{j : t_j = i_1\}| = |\{j : t_j = i_2\}|$. Recalling that $D^{t_j}_{i_1, i_2}$ is bounded, it follows from Hoeffding's inequality that

$$\Pr(|D^{\vec{t}}_{i_1, i_2}| > n^{0.6}) = \exp(-\Theta(n^{0.2}))$$

Recall that as a result of only having Type 1 BR steps, the equilibrium winner must be among the initial potential winners of any truthful profile [Reyhani and Wilson, 2012]. Therefore, for any $\vec{t} \in \mathcal{T}_3$, following the law of total probability, we have

$$\Pr\left( \text{D}^+(\vec{Q}) > n^{0.6} \mid \vec{T} = \vec{t} \right) \leq 6 \exp(-\Theta(n^{0.2})) \tag{16}$$

Combining Equations (14), (15), and (16) with Equation (13) yields our claim:

$$\begin{aligned}
\overline{\text{PoA}}(\{1,2,3\}) &= \sum_{\vec{t} \in \mathcal{T}_3} \Pr(\vec{T} = \vec{t}) \times \mathbb{E}[\text{D}^+(\vec{Q}) \mid \vec{T} = \vec{t}] \\
&\leq \sum_{\vec{t} \in \mathcal{T}_3} \Pr(\vec{T} = \vec{t}) \left[ 6n(u_1 - u_m) \exp(-\Theta(n^{0.2}))) + n^{0.6}(1 - 6\exp(-\Theta(n^{0.2}))) \right] \\
&= \Pr(\vec{T} \in \mathcal{T}_3) o(n)
\end{aligned}$$

**Step 3. Determine the probability of three-way ties.** Notice that $\Pr(\vec{T} \in \mathcal{T}_3)$ is equivalent to the probability of three-way ties under plurality w.r.t. IC, which is known to be $\Theta(n^{-1})$ [Gillett, 1977]. Alternatively, it can be proved by representing $\Pr(\vec{T} \in \mathcal{T}_3)$ as a polyhedra in $\mathbb{R}^{m!}$, which can be equivalently described by a system of linear inequalities, and then applying [Xia, 2021, Theorem 1], as in the proof of [Xia, 2021, Theorem 3]. This method can be easily extended to other cases where $\{1, 2, 3\}$ are potential winners and not exactly tied, which is not covered by previous studies on the likelihood of ties [Gillett, 1977, Xia, 2021].

For completeness, we recall from [Xia, 2021] the system of linear inequalities used to represent the winners being $W$ under any integer positional scoring rule $r_{\vec{s}}$.

**Definition 5** (**Score difference vector**). *For any scoring vector $\vec{s}$ and pair $a, b \in \mathcal{A}$, let $Score_{a,b}^{\vec{s}}$ denote the $m!$-dimensional vector indexed by rankings in $\mathcal{L}(\mathcal{A})$: $\forall R \in \mathcal{L}(\mathcal{A})$, the $R$-element of $Score_{a,b}^{\vec{s}}$ is $\vec{s}(R, a) - \vec{s}(R, b)$.*

Let $\vec{x}_{\mathcal{A}} = (x_R : R \in \mathcal{L}(\mathcal{A}))$ denote the vector of $m!$ variables, each of which represents the multiplicity of a linear order in a profile. Therefore, $Score_{a,b}^{\vec{s}} \cdot \vec{x}_{\mathcal{A}}$ represents the score difference between $a$ and $b$ in the profile whose histogram is $\vec{x}_{\mathcal{A}}$. For any $W \subseteq \mathcal{A}$, we define the polyhedron $\mathcal{H}^{\vec{s}, W}$ as follows.

**Definition 6.** *For any integer scoring vector $\vec{s}$ and any $W \subseteq \mathcal{A}$, we let $\mathbf{E}^{\vec{s}, W}$ denote the matrix whose row vectors are $\{Score_{a,b}^{\vec{s}} : a \in W, b \in W, a \neq b\}$. Let $\mathbf{S}^{\vec{s}, W}$ denote the matrix whose row vectors are $\{Score_{a,b}^{\vec{s}} : a \notin W, b \in W\}$. Let $\mathbf{A}^{\vec{s}, W} = \begin{bmatrix} \mathbf{E}^{\vec{s}, W} \\ \mathbf{S}^{\vec{s}, T} \end{bmatrix}$, $\vec{b} = (\vec{0}, -\vec{1})$, and let $\mathcal{H}^{\vec{s}, W}$ denote the corresponding polyhedron.*

For example, for $W = \{1, 2, 3\}$, $\mathcal{H}^{\vec{s}_{plu}, W}$ is represented by the following inequalities.

$$\forall \{i_1, i_2\} \subseteq [3] \text{ s.t. } i_1 \neq i_2, \sum_{R:top(R)=i_1} x_R - \sum_{R:top(R)=i_2} x_R \leq 0$$

$$\forall i_1 \in [3], i_2 \in [4, m], \sum_{R:top(R)=i_2} x_R - \sum_{R:top(R)=i_1} x_R \leq -1$$

Other cases of $\mathrm{PW}(P) = \{1, 2, 3\}$ can be characterized by modifying $\vec{b}$ accordingly. For example, $s_P(1) + 1 = s_P(2) = s_P(3)$ is represented by the following inequalities.

$$\sum_{R:top(R)=1} x_R - \sum_{R:top(R)=2} x_R \leq -1$$

$$\sum_{R:top(R)=2} x_R - \sum_{R:top(R)=1} x_R \leq 1$$

$$\sum_{R:top(R)=2} x_R - \sum_{R:top(R)=3} x_R \leq 0$$

$$\sum_{R:top(R)=3} x_R - \sum_{R:top(R)=2} x_R \leq 0$$

$$\forall i \in [4, m], \sum_{R:top(R)=i} x_R - \sum_{R:top(R)=2} x_R \leq -1$$

$\square$

# B  Experiments

Figures 4 and 5 were generated by fixing $m = 4$ alternatives with the Borda utility vector $\vec{u}_{Borda} = (3, 2, 1, 0)$, and varying the number of agents. For each $n \in \{100, 200, \ldots, 1000\}$, we sampled 10 million profiles uniformly at random and determined, for each $P \sim IC^n$, its equilibrium winning set $\mathrm{EW}(P)$. We then computed each profile's adversarial loss $D^+(P)$ and averaged their values across all profiles with the same $n$. Experiments were run on an Intel Core i7-7700 CPU running Windows with 16.0 GB of RAM.

Figure 4 demonstrates the sample average adversarial loss using these parameters. Figure 5 partitions the loss based on $\alpha$-way ties, $\alpha \in \{2, 3, 4\}$. We note the average adversarial loss decreases as

$n$ increases and takes the trend of the two-way tie case complexity. Since a significant proportion of profiles have no BR dynamics, the overall trend keeps close to zero. Therefore these results support our main theorem in this paper, that the welfare of the worst-case strategic equilibrium winner is greater than that of the truthful winner when agents' preferences are distributed according to IC.

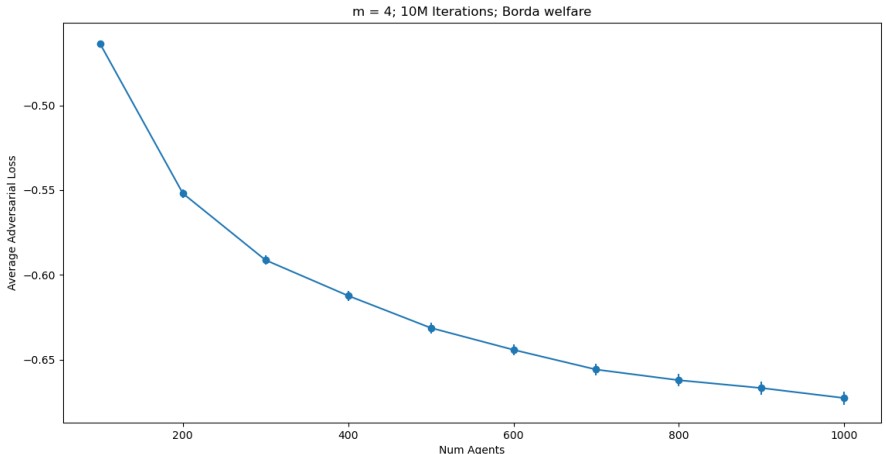

Figure 4: Average adversarial loss with $m = 4$, $\vec{u}_{Borda}$, and 10M samples. Error bars represent 95% confidence intervals, too small to see.

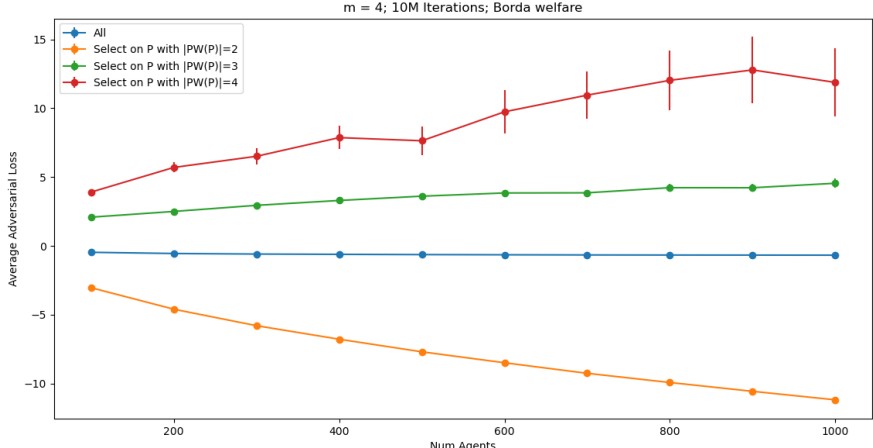

Figure 5: Average adversarial loss partitioned by $\alpha$-way ties, $\alpha \in \{2, 3, 4\}$. Error bars represent 95% confidence intervals.