# OpenReview forum: "Strategic Behavior is Bliss: Iterative Voting Improves Social Welfare"
_NeurIPS.cc/2021/Conference — NeurIPS 2021 Poster_

### Official Review · Reviewer_kFVz · 2021-07-16

**Rating:** 6
**Confidence:** 3

**Summary:**

The authors study strategic voting in the context of iterative plurality voting, in which voters may update their plurality votes over multiple rounds of voting. They study the additive dynamic price of anarchy (ADPoA), and show that in the worst-case, ADPoA can be linear in the number of agents, but for the case of impartial culture, equilibrium winners actually have a constant order advantage over the truthful winner, which is an instance of social welfare benefit due to strategic manipulation.


**Ethical Concerns:**

None.

**Limitations And Societal Impact:**

Yes.

**Main Review:**

I found this paper to be well-written and interesting -- I haven't seen many results that demonstrate that (in some settings) strategic manipulation can result in a strict increase in overall social welfare. The first theoretical result, which shows that the ADPoA can be linear in the number of agents in the worst case, was not too surprising and used relatively standard techniques, but is still a nice result that shows considerable separation from the second impartial culture result.

Speaking of that result, I would have liked to see a bit more intuition given about why this unintuitive result arises. Is it because there isn't much imbalance in impartial cultures, and allowing people to perturb their votes a bit will naturally help? I found the proof, which cases on the cardinality of W, a bit lacking in intuition here. (All I got is that the gain whenever there are two potential winners offsets the small losses whenever there are more than two potential winners.)

I also think this paper would benefit from a simulation section in which the authors could explore a range of randomized / average-case settings with mixtures that are not just uniform (impartial culture). It would be very interesting to see how robust the theoretical result for impartial culture is as the culture becomes less impartial (i.e., more imbalanced). Is the degradation in benefit smooth in some other parameter? Does strategic behavior also help when the process is simulated on (approximately) real-world data?

Minor comments:

176: applied to break it
176: suggest saying that Lemma 1 and Theorem 2 appear later in the text
Footnote 1: if one exists
255: results combine

**Time Spent Reviewing:**

2

---

> ### Author Response · Authors · 2021-08-10
> **First response to reviewer's initial comments.**
>
> Thank you for taking the time to read and review our work. We appreciate your feedback and look forward to incorporating your suggestions. Firstly, your summary and understanding of our primary contributions are correct.
>
> * (Paragraph 2) Your conclusion that the gain from profiles with two-potential winners offsets the small losses from other profiles is correct (see line 255, 324). Lemma 1 suggests that iterative plurality can “self-select” alternatives with better social welfare (see line 89). However, since the probability of a $\alpha$-way tie is decreasing ( i.e. $\Theta(n^{(1-\alpha)/2})$; see [Xia, 2021]), the expected gain in $D^+(P)$ for profiles with $\alpha \geq 3$ potential winners cannot outweigh the $\alpha=2$ case (see line 324). Therefore, the total expectation difference stays negative.
> * (Paragraph 3) We agree that further experimentation (beyond that of Appendix B) about the theory’s robustness would be insightful to the audience and think your recommendation is a natural topic for the next work (where we also hope to develop some theoretical results beyond IC). We will emphasize this discussion in the conclusion of the revision (see lines 79, 334).
> * (Minor Comment) Thank you for pointing these grammatical mistakes and typos out. These will be corrected in the revision.

---

### Official Review · Reviewer_TFbu · 2021-07-16

**Rating:** 7
**Confidence:** 3

**Summary:**

This study deals with the issue of strategic manipulations in sequential and interactive votings, and assesses how agents' welfare changes when in equilibrium.


**Limitations And Societal Impact:**

The authors clearly state the limitations of their work.

**Main Review:**

The manuscript is hard to read which penalizes the work; many sections need rephrasing to help the reader to appreciate the work.
For instance in the abstract it is stated:
"We first negatively demonstrate that the worst-case ADPoA is linear in the number of agents." ->  what negatively demostrate mean here? that there is negative linear relationship?

Please carefully proofread the document to fix typos, grammar issues, and expressivity issues for instance:
"Therefore, the positive result by Braˆnzei
55 et al. [2013] does not extend to utility functions beyond plurality utility."  as utilities you mean the other two voting rules (veto, Borda)?

Please also consider reorganizing the manuscript so that the reader has all the necessary background to follow the narrative. For instance in the theoretical background, implications, limitations and techniques could be mentioned after the literature review.

The conclusive remarks could be more insightful connecting the findings of the study to real life scenarios.

**Time Spent Reviewing:**

2

---

> ### Author Response · Authors · 2021-08-10
> **First response to reviewer's initial comments.**
>
> Thank you for taking the time to read and review our work. We appreciate your feedback and look forward to incorporating your suggestions. Firstly, your summary and understanding of our primary contributions are correct.
>
> * (Paragraph 1) We apologize for the confusion---initially we used "negatively demonstrate" to refer to the quality of the result rather than a characteristic of the result. This sentence will be modified as follows: "We first prove that the worst-case ADPoA is linear in the number of agents. To overcome this negative worst-case result, we study the average-case ADPoA and prove that equilibrium winners have a constant order expected welfare advantage over the truthful winner."
>
>
> * (Paragraph 2) We suspect that the reviewer asked whether the result by Branzei et al. is upheld if different scoring rules are used, such as veto and Borda. This is correct (see line 201), although we intended to say that the positive result by Branzei et al. (i.e., their Theorem 3) is not upheld if different utility vectors are used. To make this sentence more clear, we will modify it as follows: "Therefore, the positive result achieved by Branzei et al. [2013] is not upheld if other utility vectors are used instead of plurality utility under the plurality scoring rule."
> * (Paragraph 3) We will proofread the manuscript to improve various points of confusion, such as before Theorem 2 (line 245) and moving the Implications and Limitation section to the conclusion. See our comments for Reviewer YQpz.
> * (Paragraph 4) In addition to rearranging the introduction, we will be sure to improve the conclusion by tying in applications of iterative voting (see line 25). Moreover, we will discuss in the revision how strategic manipulations might benefit social welfare in other real-world choice settings, such as in combinatorial or division of goods domains.

---

### Official Review · Reviewer_xnBa · 2021-07-17

**Rating:** 6
**Confidence:** 3

**Summary:**

This paper studies iterative voting in which agents' utility vector could be different from the scoring vector. The authors show that PoA bounds is linear in the number of agents. Surprisingly, in an i.i.d. environment, the equilibrium winners have a constant order welfare advantage over the truthful winner.

**Limitations And Societal Impact:**

*. The comparison is against the benchmark under truthful bidding, which is generally different from the optimal benchmark when the utility vector is different from the scoring vector. Therefore, it is possible to see improvement under strategic behavior against the truthful benchmark. It would be better to highlight this fact in the fact somewhere. Is there any result regarding the approximation under strategic behavior against the optimal benchmark?

**Main Review:**

This paper is well-written, clear, and easy to follow. It is very interesting to see that in theory, when agents' utility vector could be different from the scoring vector, strategic behavior can help improve social welfare when agents are i.i.d.. However, the setting, in which agents' utility vector could be different from the scoring vector, needs further motivations and justifications; and the reviewer is curious about the social welfare performance under strategic behavior against optimal benchmark instead of the truthful benchmark.

Comments:

*. There is little motivation on when and why agents' utility vector could be different from the scoring vector. If the agents' utility vector is public, then it is unclear why the voting mechanism does not use the agents' utility vector as the scoring vector. The reviewer guesses one possible motivation is that agents' utility vector is unknown. It seems that Theorem 2 works even when the utility vector is unknown. Could you confirm?

*. When the agents' utility vector is unknown, is it possible to design a truthful voting mechanism to achieve optimal social welfare? If not, what is the best approximation one can achieve? These might be interesting questions to consider if they are not solved in the literature.

*. Theorem 1 looks a bit strange as it has no requirement on n, the number of voters. When n < 2m, the lower bound could be negative?

**Time Spent Reviewing:**

2

---

> ### Author Response · Authors · 2021-08-10
> **First response to reviewer's initial comments.**
>
> Thank you for taking the time to read and review our work. We appreciate your feedback and look forward to incorporating your suggestions. Firstly, your summary and understanding of our primary contributions are correct.
>
> * (Bullet 1) The idea that utility vectors differ from the scoring vector is due to privacy (see lines 70, 109) and infeasibility for the mechanism to attain each agent’s utility vector. We point out one challenge of prior work which assumes that the utility vector is the same as the scoring vector in lines 202-207. You are correct in that Theorem 2 works even if the utility vector is not known. We will make this clear in the revision.
>
> * (Bullet 2 and Limitations) Attaining optimal social welfare is related to work on distortion, which bounds the social welfare ratio between a voting rule and the optimal alternative (see line 331). This is a natural and good question for future work.
>
> * (Bullet 3) You are correct to suggest that the lower bound on line 218 should say “for $n>2m$.” We will include this condition in the revision.

---

### Official Review · Reviewer_YQpz · 2021-07-20

**Rating:** 7
**Confidence:** 3

**Summary:**

The authors study the welfare of Nash equilibria reached via best-response dynamics from the truthful profile. For worst-case profiles, these equilibria can have linearly less welfare than the truthful outcome. Assuming impartial culture, however, the expected welfare of these equilibria is higher than the welfare for the truthful outcome, assuming rank-based utilities.

**Limitations And Societal Impact:**

I was satisfied with how limitations and social impact were discussed.

**Main Review:**

I enjoyed reading the paper: Iterative voting seems like an important setting that we should understand better. I'm not familiar enough with prior work to know how surprising the results are, but I found the contrast between the worst-case bound (which fills in a natural gap in Branzei et al.'s analysis) and the impartial-culture bound neat. The overall presentation is clear, even though I had trouble following the proofs of Lemma 2 & 3 (I give some feedback on specifics of the presentation below). While best-responses for plurality naturally have a very restricted structure, the proofs seem to have nontrivial technical ideas.

Other comments:
- It seems like Lemma 2 includes quite clever ideas, which the reader should take away from the paper (given that the proofs are in the body). What I dearly missed while reading was a roadmap of the proof that would more clearly connect to the steps 1-4 (I found the titles of these steps rather uninformative). In line 283, it was not clear to me that the entries of the vectors were the first-ranked alternatives. Step 2 of the proof introduces multiple variables (especially the curly-Z_t,k) without giving intuitive explanations for them; this makes the proof very notationally dense and harder to follow than necessary.
- I did not quite get where the authors were going in lines 80-84. Clearly, the cost of strategic behavior is not equally distributed among agents; agents whose top-ranked alternative is the plurality winner can only lose and agents who rank the plurality winner last can only win. When the authors propose quasi-linear utilities, do they suggest to compensate election losers with money? That's an interesting suggestion, but definitely departs from the motivating scenarios and opens the gates for more elaborate manipulation. In short: I'm interested in hearing what the authors suggest here, but I don't see it yet based on their description.
- The paragraph "Implications and limitations of the main results" in Section 1.2 was hard to appreciate before seeing the content of the paper itself. To me, these points would have more naturally into a discussion at the end.
- I was quite confused by how Lemmas 2-4 are stated and proved within the proof of Theorem 2. Instead, I would recommend ending the proof at line 255, which, with some forward-referencing to the lemmas, essentially already concludes the argument.
- lines 215/216 make for a really awkward transition into Thm. 1, given that they refer to Thm. 2, and given that Thm. 1 talks about cases that the example does not generalize to well.
- typo in footnote 1: "i[f] one exists"

**Time Spent Reviewing:**

3

---

> ### Author Response · Authors · 2021-08-10
> **First response to reviewer's initial comments.**
>
> Thank you for taking the time to read and review our work. We appreciate your feedback and look forward to incorporating your suggestions. Firstly, your summary and understanding of our primary contributions are correct.
> * (Bullet 1) We intended to overview the proof of Lemma 2 in lines 269-275. Steps 1 and 2 correspond to lines 269-270 and identify which types of profiles satisfy case 1 ($\mathcal{E_1}$). Step 3 corresponds to lines 271-272, and Step 4 corresponds to lines 273-275. We will expand this high-level overview in the revision by explicitly connecting it to each step of the proof. Specifically, we will use content of the appendix (see lines 543-547, 561-564) to explain how $\mathcal{T_2}$ and $\mathcal{Z}_{\vec t, k}$ are used to represent all possible profiles that satisfy the conditions of $\mathcal{E}_1$.
> * (Bullet 2) Lines 80-84 discuss a possible line of inquiry pertaining to the fairness or equitability of iterative voting. As you indicated, agents with more polar preferences among alternatives in the potential winning set are at higher risk for losing utility. We intended to hint how fairness could be considered when agents manipulate their preferences, although this is not the primary direction of the current work. This section will be clarified and moved to the conclusion (see below).
> * (Bullet 3) We agree with your and Reviewer TFbu’s suggestion that the Implications and Limitations section should be reordered within the work to improve its clarity. Our revision would have Techniques (85-92) follow Our Contributions (47-63) in Section 1.1. Implications and Limitations (65-84) would be moved to the conclusion in order to not split up our discussion on future directions. These changes would improve the flow of the text and reduce confusion, as mentioned by Reviewer TFbu.
> * (Bullet 4) Thank you for the suggestion. Essentially that is what we are trying to do, since after line 255 we include three lemmas and line 325, which summarizes what we mentioned on line 255. The three lemmas focus on each of the cases of two, three, and four-way tied profiles. Each has different conditional expected values for the adversarial loss $D^+(P)$ (see our response to Reviewer kFVz and line 262).
> * (Bullet 5) The example on line 214 is placed to primarily clarify the adversarial loss, and to secondly foreshadow that the adversarial loss can be negative. We will add further discussion before Theorem 1 in the revision to better juxtapose the worst-case from the average-case.
> * (Bullet 6) Thank you for pointing this typo out. This will be corrected in the revision.

---

> > ### Comment · Reviewer_YQpz · 2021-08-27
> > **Post rebuttal**
> >
> > Thanks for your response, and I’m looking forward to seeing these edits in the camera-ready version! My score hasn’t changed (7).

---

### Decision · Program_Chairs · 2021-09-27

**Decision:**

Accept (Poster)

**Comment:**

Thank you for your submission. The reviewers reached a consensus that this paper is well-written and presents interesting results. Please follow the reviewers' suggestions to improve the paper in the next revision.